REGISTERED REPORT PROTOCOL

# A randomised controlled trial to test the effects of fish aggregating devices (FADs) and SBC activities promoting fish consumption in Timor-Leste: A study protocol

**Alexander Tilley**[1]*, **Kendra A. Byrd**[1], **Lauren Pincus**[1], **Katherine Klumpyan**[2], **Katherine Dobson**[2], **Joctan dos Reis Lopes**[3], **Kelvin Mashisia Shikuku**[1]

**1** WorldFish, Bayan Lepas, Penang, Malaysia, **2** Mercy Corps, Dili, Timor-Leste, **3** WorldFish Timor-Leste, Dili, Timor-Leste

\* alex.tilley@gmail.com

## Abstract

Timor-Leste is one of the world's most malnourished nations where micronutrient-deficient diets are a contributing factor to the prevalence of child stunting, currently estimated to be 45.6% of children under five. Fish are an important source of nutrients and one that may assist the country's predominantly rural population of agriculturalists to exit poverty and malnutrition. However, a small national fishing fleet producing low catch volumes places fish out of reach of most inland and upland populations where it is needed most. Fish consumption is very low in rural, inland areas compared to coastal, regional, and global averages. This study is a one-year, partially masked, cluster-randomized controlled trial among families living in rural, inland Timor-Leste. We aim to test and compare the effects of two treatments, alone and in combination, on the frequency and volume of household fish consumption in rural, inland areas as a proxy for improved dietary diversity and micronutrient intake. Treatment 1 is the installation of nearshore, moored fish aggregating devices (FADs) to improve catch rates with existing fishing gears. Treatment 2 is a social and behaviour change (SBC) activity to promote fish consumption. Villages in inland communities will be randomized to receive treatment 1, treatment 2, both treatments, or neither treatment. Data will be collected at baseline (prior to the rollout of the treatments) and endline. Our study will determine the impact of an improved supply of fish, along with nutrition-oriented SBC activities, on the fish purchasing and consumption practices of rural, inland households. Findings from this study are urgently needed by Small Island Developing States to guide policy and investment decisions on how best to improve households' diets using locally available, nutrient-dense foods such as fish. Investments such as these are needed to break the cycle of malnutrition. This trial is registered at clinicaltrials.gov (NCT04729829).

**Trial registration:** Trial registered at clinicaltrials.gov Identifier: NCT04729829.

**Data Availability Statement:** All relevant data from this study will be made available upon study completion.

**Funding:** This study is an element of the Timor-Leste Fisheries Sector Support Program - Phase 2, funded by the Royal Norwegian Embassy in Jakarta (https://www.norway.no/en/indonesia/) and led by WorldFish. The funders had and will not have a role in study design, data collection and analysis, decision to publish, or preparation of the manuscript.

**Competing interests:** The authors have declared that no competing interests exist.

## Introduction

Timor-Leste is a country afflicted by multiple burdens of malnutrition, and sustains one of the world's highest rates of stunting (low height for age), afflicting 47.1% of children under five [1]. Stunted growth is multifactorial, but in Timor-Leste, a contributing cause is likely to be a monotonous diet [2]. There is mounting evidence that fish is an underutilised resource that has the potential to improve dietary quality, but consumption of fish is low at ~6 kg/person/yr [3] compared to neighbouring countries in Asia and the global average of ~20.5 kg/person/yr [4]. However, this 6 kg average figure masks the large variation in fish consumption; coastal communities consume ~17 kg/person/year, while inland communities consume fish only seasonally, resulting in a consumption rate of ~4 kg/person/yr [3].

Fish consumption in inland and rural regions of Timor-Leste is constrained in part due to the low catch volumes landed by the small national fleet of mostly non-motorized canoes [5]. Additional barriers, such as limited road and market infrastructures in inland areas, also reduce inland consumers' access to fish. Decades of low access to marine fish in inland areas [3] has decreased consumers' expectations of consuming this food source and made them unlikely to demand it from local traders.

Fish aggregating devices (FADs) are a technology to concentrate fish to make them easier to find and catch, and there is evidence of their improving catch rates in both inshore [6, 7] and offshore settings [8, 9]. Offshore FADs can be anchored or drifting, and are used to target tuna and other highly valued pelagic fish [10, 11]. Anchored nearshore FADs are utilized by shore-based fishers to catch small pelagic fishes such as mackerels and scads [6]. By enabling access to a previously inaccessible fish stock, FADs can increase the climate adaptive capacity of fishing communities by diversifying the species caught and by reducing the dependence on production from vulnerable reef habitat [12].

Investigating the barriers for fish from small-scale fishers to reach the populations in need is important for understanding the contribution of fisheries to nutrition security. Proximity to inland fisheries is associated with an increase in fish consumption in children in rural Sub-Saharan Africa [13]; however, more information is needed to determine if this relationship holds in other contexts. Fish aggregating devices can increase supply, but increased availability alone may not be adequate to increase consumption, especially when it comes to using animal-source foods for infant and child feeding [14]. Building consumers' awareness and demand for high-quality marine fish must go together with increasing fish supply to improve the contribution of fisheries to nutrition security in inland areas.

Beyond increasing supply and awareness on benefits of fish consumption, a woman's decision-making power, sociocultural practices and seasonality have strong impacts on a household's ability to consume fish. A Gender Equality and Social Inclusion Analysis (GESIA) conducted in 2017 [15] found that a major gender equity issue in nutrition relates to the constraints that women face in decision-making and their lack of control over food, which impacts significantly on nutrition choices and the prevention of malnutrition. Research has also found that mothers (younger mothers in particular) were not confident to spend money on small quantities of meat or canned fish without permission from their husband [15, 16].

Promoting a healthy diet and optimal infant and child feeding practices through contextualized social behaviour change (SBC) interventions is another avenue to improve dietary quality and eventual health outcomes. The influence of targeted SBC approaches on diets and expenditures is mixed [17, 18], however, some programs have shown success [19]. A key component of successful interventions is a deep understanding of the community in which the SBC intervention is taking place, and inclusion of local perspectives and beliefs [20]. Mercy Corps, an international NGO, has been working in Timor-Leste since 2007, and has provided

interventions that are compatible with its ethnography [21]. Mercy Corps provides rural households with support through a sequenced layering approach. The sequence begins with financial services through Village Savings and Loans Associations (VSLAs). With access to secure savings, a source of credit, and an opportunity to earn interest on their investment, VSLAs provide a foundation for the promotion of key nutrition practices through participatory discussions within established VSLA members [22]. Thus, our study design will allow us to evaluate the feasibility of using VSLAs with contextually appropriate nutrition discussions to improve dietary quality. Furthermore, this study will provide robust evidence to determine the impact of both supply and demand interventions, alone and in combination, to increase fish consumption and improve dietary quality in rural, inland Timor-Leste.

## Design and methods

**Objectives and hypotheses.** Our trial aims to evaluate the effectiveness of increased fish production and targeted promotion of the nutritional benefits of fish consumption through social behaviour change (SBC) activities on the frequency and quantity of fish consumption in rural households in Timor-Leste. Findings from our study will inform governments, practitioners, and donors on the implementation of effective interventions to improve household diets through increased fish consumption. Our hypotheses are as follows:

$H_0$ Deployment of FADs has no effect on volume and frequency of fish consumed

$H_0$ Providing SBC has no effect on volume and frequency of fish consumed

$H_0$ Combined deployment of FADs and SBC has no effect on volume and frequency of fish consumed

$H_0$ Deployment of FADs and SBC have no differential impacts on consumption

Our study is a one year-long, cluster randomised controlled trial (RCT) among rural, inland households using a superiority framework to test the effect of the interventions, but alone and in combination. We will employ a parallel intervention (Fig 1), using a 4:2 treatment to control ratio at the municipality level (FADs vs no FADs), and a 1:1 ratio among intervention and control villages (SBC vs no SBC).

## Study setting

This study will be carried out in six coastal municipalities of Timor-Leste (Bobonaro, Dili, Liquica, Manufahi, Manatuto and Covalima) (Fig 2). The rural, inland areas of these municipalities exhibit some of the highest rates of child stunting in the world at 47.1% [23].

## Supply side intervention

**Inclusion/exclusion criteria of villages selected to receive FADs.** On the supply side, the experiment will involve the deployment of two fish aggregating devices in the nearshore fishing grounds of each coastal fishing village with already established trading links to inland study sites. Using WorldFish and government data on fisheries landing sites in the six municipalities, rural coastal fishing villages with over 10 active fishing vessels will be listed. Villages that already have or utilise FADs will be excluded. Six villages (one per municipality) will then be randomly assigned to one of two RCT arms, namely FAD (4 villages) and non-FAD (2 villages) (Fig 3). This treatment arm is imbalanced 2:1 because although the efficacy of nearshore FADs at increasing catch rates of fish has been established [6], it is also shown to be dependent upon local ecological and bathymetric conditions. These coastal villages are geographically dispersed to minimize the risk of contamination between treatment and control sites. In addition, by mapping the value chains of traders in baseline focus groups, we will minimise contamination of non-FAD sites with fish caught at a FAD.

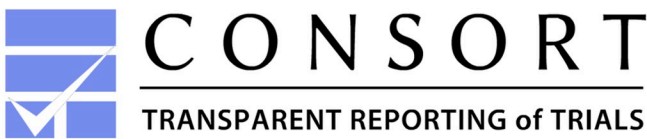

## CONSORT 2010 Flow Diagram

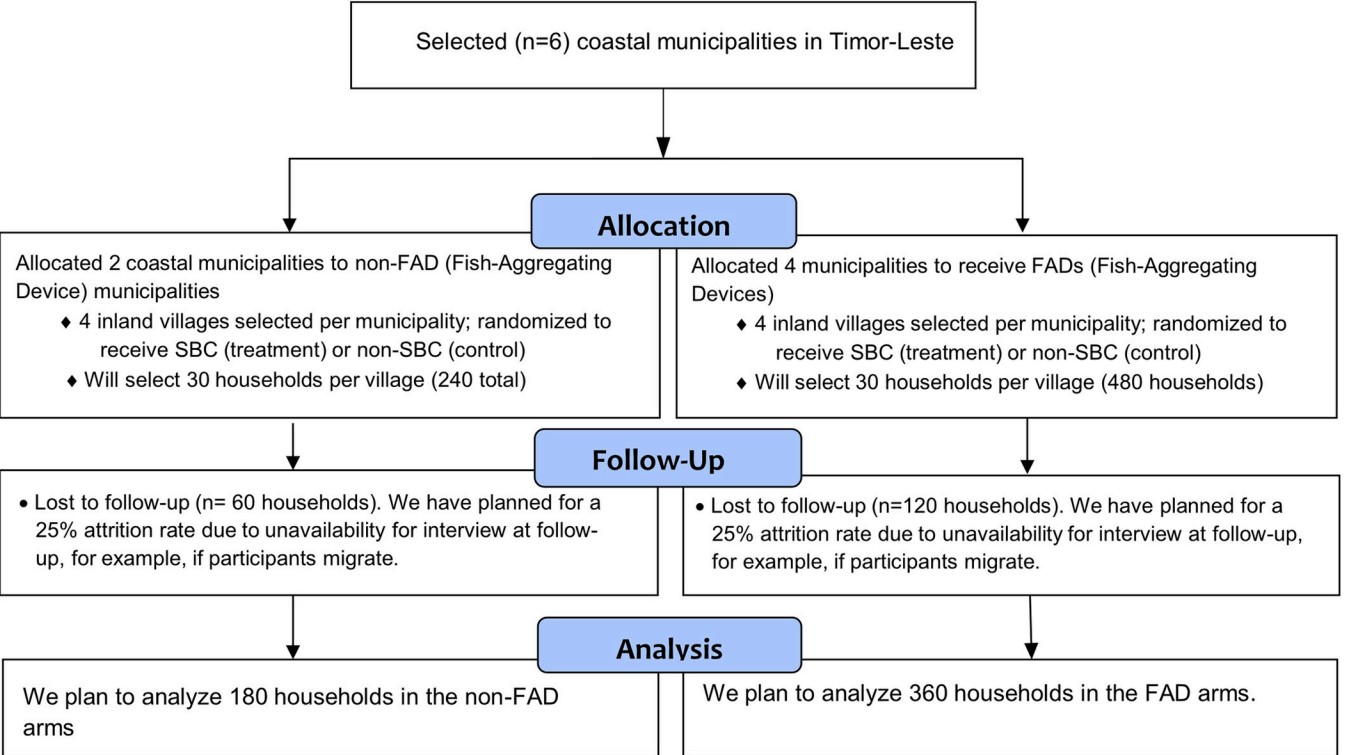

**Fig 1. A CONSORT flow diagram of the progress through the phases of the parallel randomised trial.**

### Description of the FAD intervention

The design, construction and deployment of nearshore FADs used in this study have been tested since 2013 in Timor-Leste [24]. FADs are made up of a series of buoys and attractant netting attached to nylon (sinking) and polypropylene (floating) ropes that are moored to the seabed using concrete blocks and a grapple anchor (Fig 4).

### Demand side intervention

**Inclusion/Exclusion criteria of SBC villages.**   For the SBC intervention, 4 inland villages within a 30 km radius (inland) of the coastal sites that have a VSLA established by Mercy Corps will be selected. A further inclusion criterion for the villages is that fish traders from the coastal landing sites confirm through interview that they currently sell fish products to those respective villages. These villages will then be randomised into an SBC arm (treatment) and a non-SBC arm (control). Villages will be randomly allocated to treatment or control arms using an Excel random number table to a 1:1 allocation ratio. The principal researcher will generate the allocation sequence, and the allocation of the village to either treatment or control will be

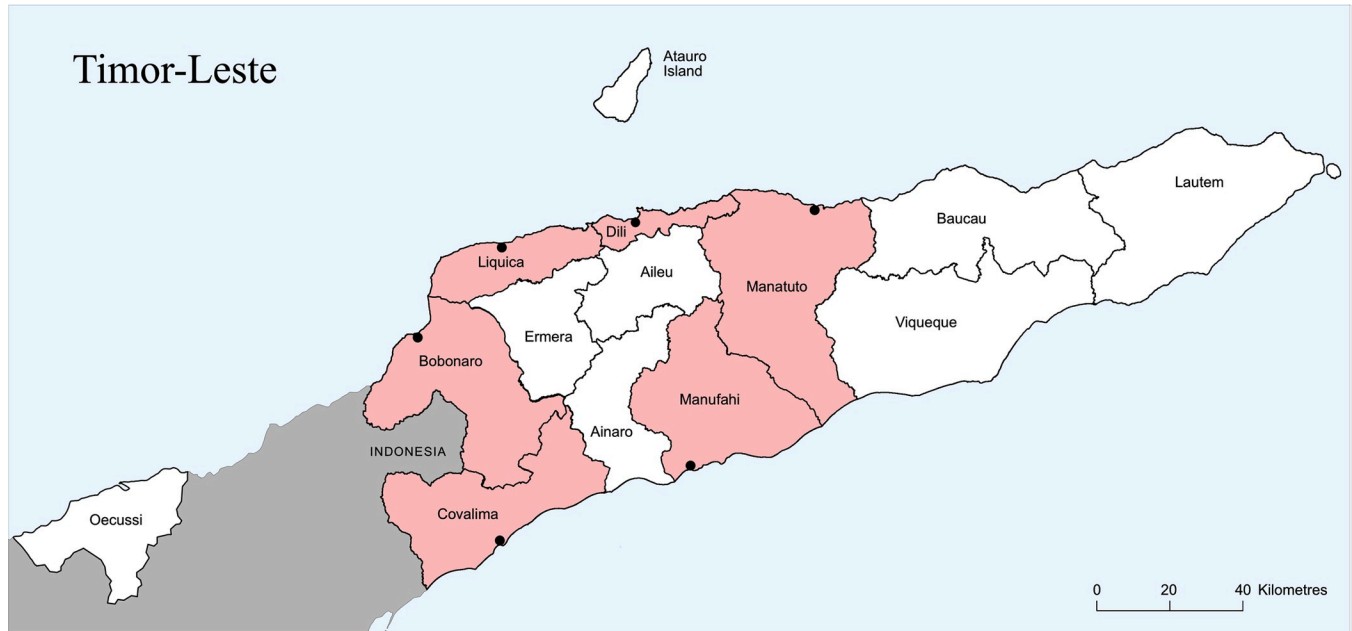

**Fig 2. Map of Timor-Leste showing municipalities.** Red shading shows the municipalities represented by coastal producer sites. Map data adapted from © OpenStreetMap contributors.

communicated to Mercy Corps, who will provide the SBC intervention to the treatment villages.

## Description of the SBC intervention

Within each SBC village, Mercy Corps will layer nutrition content and learning through VSLAs. The VSLA members will participate in skill-building activities that promote five key

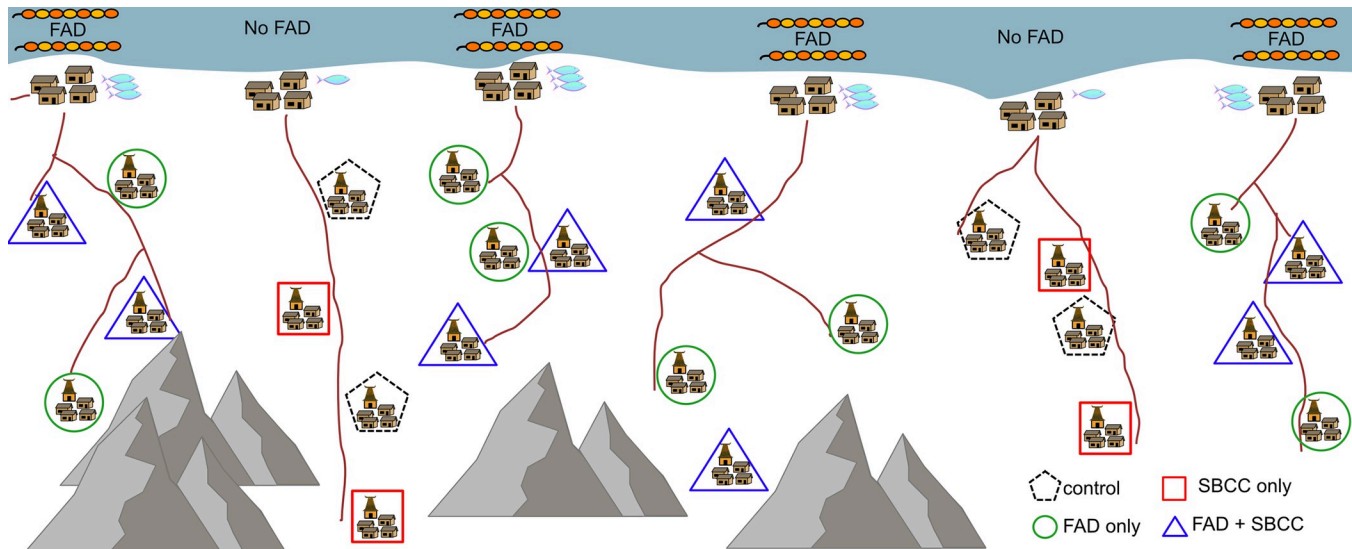

**Fig 3. A sketch representation of the two treatment levels of the randomised controlled trial.** 1. Coastal nearshore fish aggregating devices and 2. Social and behaviour change activities in rural inland communities in Timor-Leste. This diagram is a visualization only and does not represent the location of villages in the study. Diagram and icons ©A.Tilley (created using Affinity Designer software (https://affinity.serif.com/en-us/designer/).

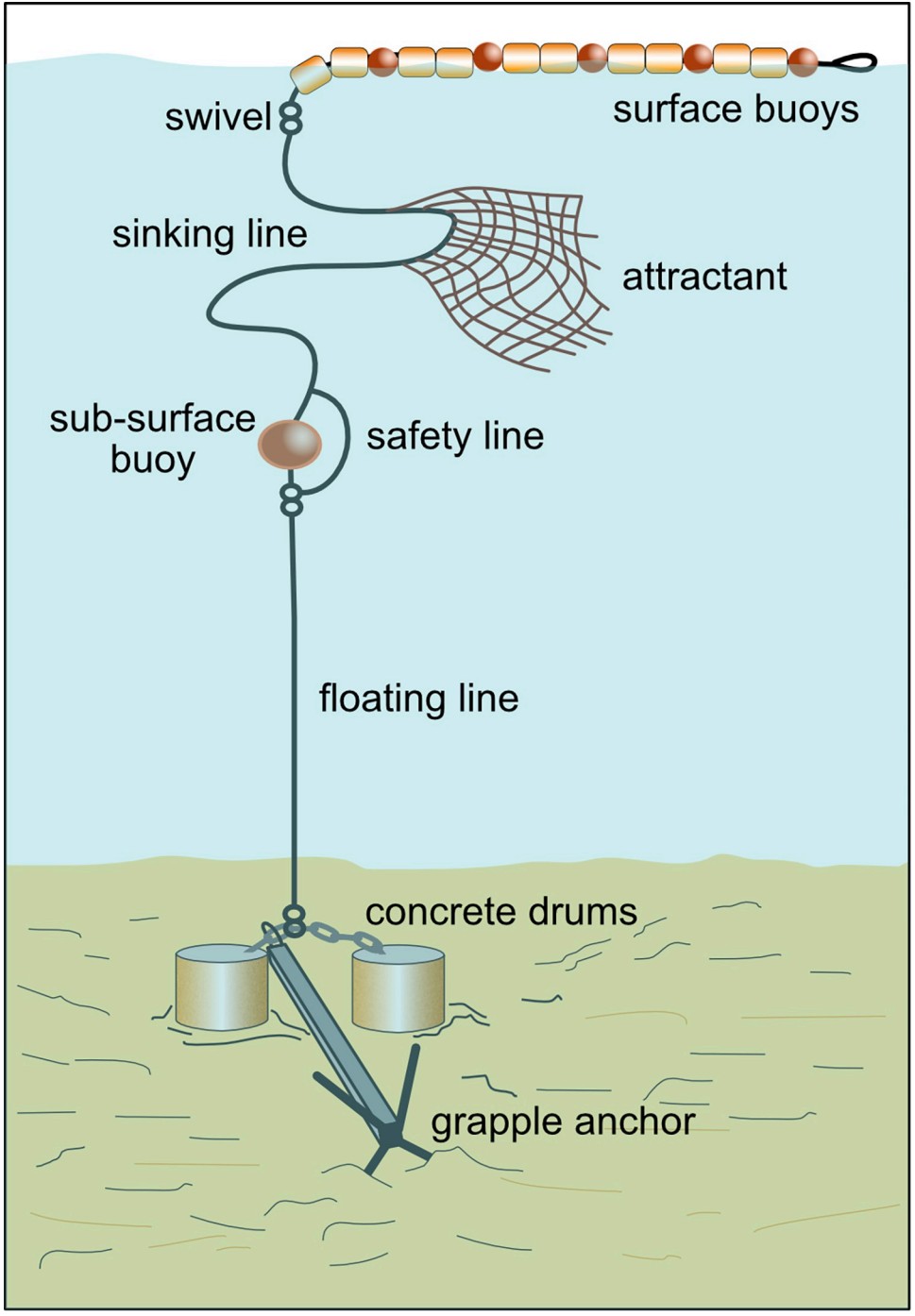

**Fig 4. A diagram of a nearshore fish aggregating device (FAD) (not to scale).** Diagram and icons © A.Tilley (created using Affinity Designer software (https://affinity.serif.com/en-us/designer/)).

behaviours about fish nutrition and household decision making on fish consumption as part of facilitated discussions, interactive learning sessions, and a video-based facilitated dialogue. Key promoted behaviours will include: 1) Incorporate fish into family meals at least twice a week; 2) Parents pick bones out of fish for small children and start offering fish to infants at 6

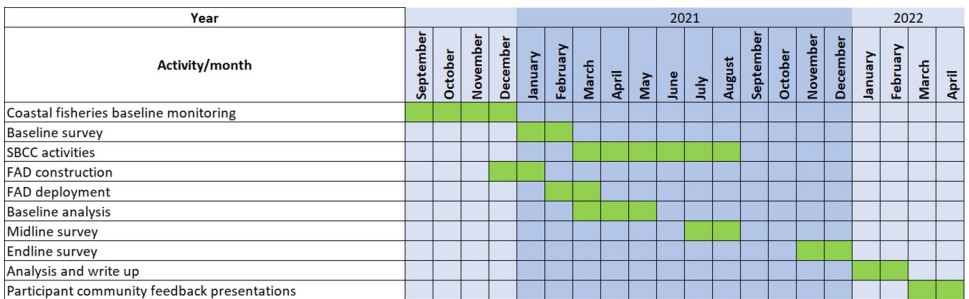

| Year | | | | | 2021 | | | | | | | | | | | | 2022 | | | |
|---|---|---|---|---|---|---|---|---|---|---|---|---|---|---|---|---|---|---|---|---|
| Activity/month | September | October | November | December | January | February | March | April | May | June | July | August | September | October | November | December | January | February | March | April |
| Coastal fisheries baseline monitoring | | | | | | | | | | | | | | | | | | | | |
| Baseline survey | | | | | | | | | | | | | | | | | | | | |
| SBCC activities | | | | | | | | | | | | | | | | | | | | |
| FAD construction | | | | | | | | | | | | | | | | | | | | |
| FAD deployment | | | | | | | | | | | | | | | | | | | | |
| Baseline analysis | | | | | | | | | | | | | | | | | | | | |
| Midline survey | | | | | | | | | | | | | | | | | | | | |
| Endline survey | | | | | | | | | | | | | | | | | | | | |
| Analysis and write up | | | | | | | | | | | | | | | | | | | | |
| Participant community feedback presentations | | | | | | | | | | | | | | | | | | | | |

**Fig 5. Timeline of interventions and data collection in the study.**

months of age; 3) VSLA members save with a purpose of protein consumption for their families, with a focus on fish consumption; 4) Couples initiate conversations on a weekly basis on allocation of resources for weekly protein purchase, inclusive of fish; 5) Households use resources to purchase fish two times a week. At the village level (N = 12), Mercy Corps will promote increased fish consumption through SBC mass attendance events through marketing of fish vendors, interactive learning, and *edu-tainment* competitions promoting fish consumption. SBC will raise awareness of the benefits of eating fish as part of a balanced diet, with a particular focus on pregnant and lactating mothers, and children under 5 years of age. The non-SBC communities will not receive the nutrition campaign package. Villages will be exposed to the SBC intervention for six months (Fig 5).

## Sample size and sampling procedure

The sampling frame comprises rural, inland households in Timor Leste. We conducted a power calculation using the command *clustersampsi* in STATA based on secondary data, and a primary outcome of fish consumption per household. We assumed a minimum detectable effect of 15% and an intra-cluster correlation coefficient equal to 0.05. To achieve at least 80% statistical power, we require a sample size of 575 inland community households selected from 25 communities (that is, we sample 23 households in each community). Adjusting for 25% attrition, we estimate that a sample size of 720 is required—about 30 households per site.

Sampling follows a multi-stage procedure. In the first stage, 6 coastal municipalities were purposively selected based on the coverage of Mercy Corps programming of VSLA groups. The second step involves compiling a list of rural coastal fishing villages with over 10 active fishing vessels and randomly selecting one village within each municipality. In the third step, a list of inland villages within a 30 km radius of each coastal villages and with active presence of a VSLA was compiled. Although we include only villages participating in VSLAs, the typical size of a VSLA (about 15 members) is less than the required number of households per village. In the fourth step, therefore, we generate a list of all households in the village, both VSLA members and non-members. To ensure the required sample size is satisfied (in accordance with the power analysis) we will randomly select additional households from the list of those not participating in VSLAs. The chosen households will be visited and asked if they would be willing to participate in the study. Where households are not willing, the closest neighbouring household will be asked, and so on.

## Recruitment and consent

Communities and study households will be informed of the study and sensitized by the research team to its activities and purpose through community meetings and meetings with

local leaders. Treatment and control groups will not be informed of their assigned group to avoid the possibility of introducing Hawthorne or John Henry effects [25]. An informed consent including maintenance of the confidentiality of personal data, and the possibility to refuse the consent without having to justify the refusal, will be obtained from all community leaders and study participants prior to responding to any surveys. Consent will be obtained by the study enumerators.

## Data collection

**FAD fisheries catch and effort data.** At each of the six coastal sites, a baseline of fishing activities will be collected from participating fishers including trip frequency, duration, method, location, and substrate type, along with the catch volume. Data will be collected from fishers as they return to shore by on-site enumerators, consistent with the national fisheries monitoring system in Timor-Leste, PeskAAS (following [26]). This is a tablet-based survey where government enumerators collect and upload near-real-time landings data to the central database. All participation in vessel tracking and contribution of data is voluntary.

Information collected includes the number of fishers (sex-disaggregated), boat and gear types, individual sizes and quantities of fish captured, trip duration, the primary habitat fished, and the proportion and price of fish sold. These data are also paired with geospatial vessel tracks for boats that are fitted with a GPS tracker. Fishing activities and catch monitoring will be established prior to installation of any FADs, to evaluate any change in catch rates or production volume following the FAD deployment. Enumerators at each of the six coastal sites will be responsible for daily data collection at landing sites of small-scale fishers and will collect data on fish catch by the species' family group.

## Household surveys

Following enrolment and informed consent, a baseline survey will be conducted with one member per household. The baseline survey will consist of household demographic information, asset ownership, and food consumption questions to assess relative wealth and standard of living, access to fish and other animal source foods, food insecurity, and the frequency and volume of fish consumed at the household level. Qualitative 24-hour recalls will be done with the household member responsible for food preparation. If there are children under five years old in the household and the respondent is not the primary caregiver, the caregiver will be asked 24-hour recall questions related to the diet of the child(ren). The 24-hour recalls will be used to calculate the minimum dietary diversity for both the woman and child [27, 28]. We will also ascertain if there are any cultural or traditional beliefs that influence fish consumption for specific people or at certain times.

Interviews will be conducted by enumerators extensively trained on the tools, methods, research ethics and confidentiality. Questionnaires will be translated into Tetum and carried out in Tetum or a combination of Tetum and other local languages. After enumerator training and survey pre-testing, the survey tools and procedures will be adapted as necessary to account for errors in interpretation by enumerators and respondents. Staff will inform the village chief and VSLA members in advance of the baseline and endline data collection sessions, and the latter will arrange for participants to be present.

## Data management

All data will be kept confidential. Data will be submitted daily via 3G for review by in-country researchers and only stored on password protected mobile phone tablets and laptops temporarily. The household survey questionnaire will include skip logics and field specifications and

delimiters to minimise input errors, but also supervising researchers will flag any unusual data for checking with enumerators. All variables will be visualized and checked for outliers. Outliers will be closely inspected, and sensitivity analyses will be conducted on high leverage data points. Outcome variables will be visually checked for normality and transformed if necessary. Participants will be assigned randomly generated ID numbers, and names and identifying information will be de-linked from the dataset. Data will be analysed for groups and no individual response will be identifiable. Identifiable survey data will be archived and accessible to only the research team and those approved by the entire research team. Anonymised data from the study baseline survey will be published on the Harvard Dataverse (https://doi.org/10.7910/DVN/VZJYIZ) to facilitate future research such as pooled analysis with other studies for purposes of meta-analysis or multi-site analysis. Anonymous fisheries landings data are publicly available on Harvard Dataverse (https://dataverse.harvard.edu/dataverse/peskas) and updated automatically every month.

## Statistical analysis

**Main outcome variables.**   The main outcomes of interest are:

1. *Per capita household fish consumption* measured as total quantity (g) of fish consumed by the household in the previous seven days divided by the number adult equivalents in the household.

2. *Frequency of fish consumed at household level* measured as the number of days fish were consumed out of the previous seven days.

## Other outcome variables

1. *Frequency of fish consumption in women* measured as the number of days fish were consumed out of the previous seven days.

2. *Frequency of fish consumption in children* measured as the number of days fish were consumed out of the previous seven days.

3. *Average catch rates* calculated as the increase in fish availability by comparing average catch rates prior to FAD deployment with catch rates post-deployment. Catch rate and total production values for different coastal community sites will be calculated and extrapolated following Tilley et al. [26].

4. *Average price and supply of fish* calculated as the change in price and volume of fish available in upland village sites by comparing price and volume of fish proffered by traders at baseline with volumes at midline and endline.

5. *Knowledge and practices about fish purchase and handling.* A series of questions assessing knowledge of nutritional benefits of fish and handling practices will be administered to respondents at baseline, midline and endline.

**Treatment variable.**   The main treatment variable is a dummy variable that indicates to which of the four treatment arms a village was assigned, but analysis will be done by a statistician masked to the treatment arms. Our primary model is an unadjusted model.

However, if we find that after unmasking, there are variables in the dataset that appear different by treatment arm, we may consider adjusting for confounders in the model.

**Mediation analysis.** We will assess whether the effect of SBC is mediated by an exogenous exposure of coastal communities to the FADs. Therefore, we will construct an interaction of the SBC treatment dummy with a dummy variable equal to one if an inland community is linked to a coastal community with access to a FAD and zero otherwise.

Additionally, we will assess whether increased fish consumption is mediated by a change in self-reported knowledge. We will do this by constructing a dummy composite variable of all knowledge questions and looking for an interaction by knowledge and treatment arm on per capita fish intakes.

**Checking for spillover effects.** In addition to our research design feature for controlling spillovers, we will formally test for evidence of spillovers across neighbouring villages using Global Positioning System (GPS) coordinates of the households. We test whether the presence of an individual from another experimental arm in a neighbouring village affects fish consumption status of the neighbour. To check whether fish consumption status of the control group neighbours were affected by spillovers, we will compare outcomes of control group neighbours who are close to a treated neighbour and control group neighbours further away from treated units. Using a border-to-treatment dummy variable, a *t*-test will be conducted to check that control group neighbours' fish consumption status were not significantly affected by the presence of a neighbour from another experimental arm.

SUTVA may also be violated if other organizations working in the study sites implement interventions similar to the ones proposed in the study. This is especially problematic if the control group is targeted because of the feeling that they have been left out of a potentially beneficial intervention. We have controlled for this potential problem by collecting data about all organizations working in the study site and the interventions implemented. In addition to the randomization, which helps to ensure both treatment and control group participants are equally exposed to the interventions of such organizations at baseline, we discussed with partner organizations and understood their planned future interventions. These do not relate to the interventions implemented in our study. Furthermore, we collaborate with Mercy Corps as our implementation partner to ensure alignment in project interventions and to avoid parallel interventions being targeted at the same respondents by different organizations.

**Dealing with attrition.** Our study design incorporates strategies to minimize attrition including working with existing VSLA groups. However, we cannot rule out that attrition may still occur. Therefore, in case of substantial attrition, the reasons will be carefully recorded. This will help us to assess whether potential causes of attrition are systematically linked to specific treatments. We will check whether attrition rates are equal across the three experimental arms or if concentrated in one arm. High attrition is potentially problematic, as it could introduce selection bias in our randomised experiment. In case of high attrition, we will examine the implication of attrition for our results in several ways. First, we will test whether attrition is affected by treatment assignment. If this is the case, we will implement Lee bounds to estimate lower and upper bounds of treatment effects [29]. Second, we will test whether our remaining sample is (still) balanced along key observable dimensions, and whether attrition is driven by different characteristics across treatment groups. To do this, a probit regression will be used where we regress attrition status on the treatment dummies, household characteristics, and the interaction of these. The hypothesis that attrition is not differentially determined by treatment status will be assessed through an F-test for the joint significance of the interaction terms in this regression. If differential determinants of attrition are observed, we will attempt to control for this through a weighting procedure as a robustness analysis [30, 31]. Specifically, we will follow a two-stage procedure. In the first stage, a logit regression is used to estimate the predicted probability of having non-missing measures for our outcomes given treatment assignment and a vector of observable covariates. In the second stage, we will weight each

**Table 1. Description of study arms and their exposures.**

| Arm | Exposures |
| --- | --- |
| FAD + SBC | • Mercy Corps VSLA site<br>• Within 30 km of a FAD<br>• SBC activities (nutrition messaging and skill building on fish consumption to improve dietary diversity)<br>• One or more retailers who provides the village with fish |
| FAD only | • Mercy Corps VSLA site<br>• Within 30 km of a FAD<br>• No SBC activities<br>• One or more retailers who provides the village with fish |
| SBC only | • Mercy Corps VSLA site<br>• SBC activities (nutrition messaging and skill building on fish consumption to improve dietary diversity)<br>• One or more retailers who provides the village with fish |
| No FAD, No SBC (control) | • Mercy Corps VSLA site<br>• No FAD within 30 km<br>• No SBC activities<br>• One or more retailers who provides the village with fish |

observation using the inverse of the thus estimated probability of having a non-missing measure of our outcomes. We will check whether our main results remain robust to all these robustness tests.

**Multiple hypothesis testing.** Because we are making inference on many hypotheses, it is possible that significant results emerge from our analysis due to chance rather than actual treatment effects. We will follow [32] and adjust the *p*-values using a number of different methods. We will calculate Romano-Wolf adjusted *p*-values (following [33]) to correct for the familywise error rate (FWER), the probability of making at least one false discovery among a family of comparisons. We will also calculate sharpened *q*-values (as in [34]) to correct for the false discovery rate (FDR), the probability of making at least one false discovery among the discoveries already made.

**Balance at baseline.** The goal of the randomization is to ensure that the treatment and control groups are similar in terms of average characteristics at baseline. This will be tested formally through an F-test of joint orthogonality using a multinomial logit regression, which tests whether the observable characteristics in Table 1 are jointly unrelated to treatment status [35]. Failure to reject this null hypothesis indicates that the randomization succeeded in achieving balance across the experimental arms. In addition, we will calculate the standardized difference in means [36, 37]. We will check whether the standardized difference in means are below the threshold of 0.25 as recommended in literature [37, 38], indicating balance.

### Empirical estimation

*Main estimation approach.* Intent-to-treat (ITT) estimates will be presented following Eq 1:

$$y_{iv} = \alpha + \sum_{k=1}^{3}\beta_k treat_v^k + \gamma_i W_{iv} + C_c + \varepsilon_{ic} \tag{1}$$

where $y_{iv}$ is the outcome variable for household *i* in village *v* (Per capita household fish consumption, frequency of fish consumed at household level, frequency of fish consumption in women, frequency of fish consumption in children, knowledge about nutrition benefits of fish). The variable $treat_v^k$ denotes the treatment dummy variables. $W_{ijm}$ is a vector of control variables including household characteristics (see Panel B in Table 2) and baseline levels of outcome variables, $C_c$ captures municipality fixed effects, and $\varepsilon_{ijm}$ is the usual idiosyncratic

**Table 2. Baseline characteristics by experiment arm.**

| Variable | SBC only | FAD only | SBC + FAD | Control (no SBC & no FAD) |
|---|---|---|---|---|
| Panel A: *Outcomes* | | | | |
| Per capita household fish consumption | | | | |
| Frequency of fish consumed at household level | | | | |
| Frequency of fish consumption in women | | | | |
| Frequency of fish consumption in children | | | | |
| Knowledge about nutrition benefits of fish | | | | |
| Dietary diversity | | | | |
| Panel B: *Other variables* | | | | |
| Age of respondent | | | | |
| Sex of respondent | | | | |
| Education of respondent | | | | |
| Household size | | | | |
| Number of women of reproductive age (15–49) | | | | |
| Number of infants | | | | |
| Per capita household income | | | | |
| Food insecurity | | | | |
| Asset score | | | | |
| Participation in other community projects | | | | |
| Number of nutrition information exchange links | | | | |

error term, clustered at the village level (the unit of randomization). Here the parameter of interest is $\beta_k$, the average treatment effect. Equation will be estimated using OLS. We will first estimate a parsimonious model with only the treatment dummies and the municipality fixed effects. Then we will control for additional baseline covariates. The control group in Eq 1 is the no-FAD and no-SBC group. Eq 1 will be estimated separately for midline and endline subsamples. However, we will compare the magnitude of impacts at midline and endline.

**Ethical and safety considerations.** Ethics approval was received from the Timor-Leste Instituto Nasionau do Saude (National Institute of Health) in December 2020 (1934MS-INS/ DE/ /XII/2020). Written informed consent will be obtained from all VSLA members and respondents before recruitment into the study. Based on the experience of similar surveys and trials, no harm is expected from trial participation. Any changes to the protocol will be reported to both the Timor-Leste Instituto Nasionau do Saude (National Institute of Health) and clinicaltrials.gov.

**COVID-19.** Fieldwork will be conducting according to Government of Timor-Leste directives regarding the Standard Operating Procedure to minimise transmission of COVID19. Precautions taken will include conducting interviews outside, at a distance not less than 2 metres from respondents and others, and with interviewer and respondent wearing masks.

## Discussion

The development and comparison of new technologies and approaches for reducing malnutrition in Timor-Leste responds to an urgent need captured in the country's current ranking at the very bottom of the global hunger index [39]. The Timor-Leste Strategic Development Plan [40], stated the goal of doubling capture fisheries productivity (an increase of 6,500 t) by 2020 to harness more animal-source foods from fisheries and aquaculture, yet to do so the government must have robust scientific evidence on which to base investment decisions. Fisheries play an important role in nutrition and livelihoods for coastal dwellers in Timor-Leste [41],

who consume almost three times more fish than the national average [3]. Yet, there remains a lack of economic incentives to invest in the sector due to poor infrastructure and low economic returns [42].

There is convincing evidence that since their adoption in the Pacific in the late 1970s, FADs have in many instances substantially increased SSF catch rates [43–45]. As such many Pacific nations have integrated FAD programs into their national fisheries action plans and policy [8, 46]. However, there is very little empirical research on the contribution of FADs to improved small-scale fisheries livelihoods and nutrition. Tilley et al. (2019) [6] established that nearshore FADs effect on catch rates was highly dependent on site ecology, but on average FADs provided a return on investment in five months or less. Furthermore, the oceanographic characteristics of Timor-Leste (steep slopes and strong currents) limit the longevity of deployed FADs, so one year is sufficient time to see any changes in catch rates.

Our study seeks to understand if increased fish volume in coastal areas is absorbed by current markets or represents sufficient surplus to drive greater supply into inland areas. It is designed to approach the push and pull market dynamic from either end, without directly incentivising actors in the market chain. By increasing the volume of fish available for sale, we are pushing surplus supply into new and smaller markets; and by encouraging fish consumption through a contextualized SBC intervention, we aim to increase the demand for fish to pull more into rural market. We do not anticipate large effects on the market price of fish in the study timeline, but we will record fisher sale price as part of monitoring. The reason for not incentivising traders to go to specific study villages is the lack of a reliable mechanism through which this could be sustained beyond the project life cycle. Individual variation between traders and their extant connections with inland communities is likely to be an important factor in decisions over where to direct greater fish production. We aim to mitigate this as far as possible through selection of traders for baseline and endline focus groups that already trade in the study inland communities.

While many people are aware of the importance of animal-source foods in Timor-Leste, and in particular the importance of fish [3], many women and children suffer from in adequate diets [47]. This study, utilizing robust evidence from interventions both alone and in combination, will generate important knowledge on household barriers to consuming more fish. In particular, our study will determine if additional encouragement to consume fish, when combined with the resources to purchase fish (via a Village Savings and Loan Association) is effective in increasing fish consumption, even without increased supply. As previously stated, there are other factors that can impact fish consumption, including seasonality, fish price, and decision-making power over household income expenditure and cultural practices. Alternatively, we may see that increased supply alone, in combination is additional household spending money is enough to increase fish consumption, even absent of an SBC campaign. These sets of interventions have not been tested in this context before, and the evidence will thus provide important information to policy-makers and stakeholders working to improved nutrition in small island developing states.

## Limitations

There are two limitations of this trial. First, the dietary data on fish consumption is based on recall and self-reporting, and therefore risks response bias. It is expected that response bias would favour over-reporting of consumption, which may suggest lower effectiveness of the interventions. Sharing of foods with family members would likely be under-reported, again leading to underestimation of effectiveness. Collection of data on potential confounders / modifiers of effect.

While gender and power dynamics are recognised as important influencers of dietary choice and diversity, the intersection of these factors with increased supply and knowledge of nutrition benefits of fish will not be explored in this study. However, VSLAs are composed of both men and women members and SBC activities will involve all members and encourage collaborative food purchasing decision-making.

The price of fish is a potential confounder in the study; however, we will collect market price and spending data on fish and other animal source foods. The market price variation of fish across sites (e.g. based on the geographical distance between producer and consumer) may affect access and thereby influence dietary preference. Information on prices and spending at each of the sites will be routinely collected as part of this study, and if variation occurs, will be included as a covariate in our adjusted models. In the same way, variable fish quality and safety between sites (fish deteriorating further with longer exposure to heat in reaching further sites) may affect consumer dietary preference. In the baseline and endline surveys, we will ask if fish quality is a concern or constraint to purchasing behaviour at each site. Travel time will be established to each site under normal conditions, and temperature loggers will also be used to track the temperature inside coolers carried by traders from producer to consumer sites.

## Conclusion

From a programmatic point of view, if FADs and SBC prove successful at increasing rates and volume of fish consumption, not only would it provide an acceptable and scalable program, but it might also highlight where other technological innovations could be paired with SBC to drive sustainable market effects without the need for costly incentives.

This study will contribute broadly to the body of knowledge on fish consumption patterns and dietary choice by the rural poor in Timor-Leste, and due to close collaboration by WorldFish and Mercy Corps with the Ministry of Agriculture and Fisheries and the FAO, our findings will be used to inform decision making around new interventions, policies, and investments in Timor-Leste. The effects of FADs on the supply of fish to inland areas will further highlight possible opportunities for using this technology to achieve national development priority objectives around nutrition and rural development. Furthermore, new knowledge generated about the effects of SBC on dietary choice with specific relevance to fish are of broad interest to those studying food systems, and the dynamics of poverty in small-island developing states and beyond.

## Supporting information

**S1 Appendix. SPIRIT checklist.**
(DOC)

**S2 Appendix. Ethical approval from the National Institute of Health, Timor-Leste.**
(PDF)

## Acknowledgments

AT, KAB, LP, and KMS designed the study and statistical analysis plan. KK, KD, KAB, and JRL developed the social behaviour change intervention program. All co-authors contributed to the writing of the manuscript.

## Author Contributions

**Conceptualization:** Alexander Tilley, Kendra A. Byrd, Lauren Pincus, Kelvin Mashisia Shikuku.

**Data curation:** Joctan dos Reis Lopes.

**Formal analysis:** Alexander Tilley, Kelvin Mashisia Shikuku.

**Funding acquisition:** Alexander Tilley.

**Investigation:** Alexander Tilley, Joctan dos Reis Lopes.

**Methodology:** Alexander Tilley, Kendra A. Byrd, Lauren Pincus, Katherine Klumpyan, Katherine Dobson, Joctan dos Reis Lopes, Kelvin Mashisia Shikuku.

**Project administration:** Alexander Tilley, Katherine Klumpyan, Katherine Dobson.

**Supervision:** Alexander Tilley, Kendra A. Byrd, Katherine Klumpyan, Joctan dos Reis Lopes.

**Validation:** Joctan dos Reis Lopes, Kelvin Mashisia Shikuku.

**Writing – original draft:** Alexander Tilley, Kendra A. Byrd, Lauren Pincus, Katherine Dobson, Kelvin Mashisia Shikuku.

**Writing – review & editing:** Alexander Tilley.

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
