## [Decision Letter · Decision Letter 0]

26 Oct 2021

PONE-D-20-40710A randomised controlled trial to test the effects of fish aggregating devices and SBC activities promoting fish consumption in Timor-Leste: A study protocolPLOS ONE

Dear Dr. Tilley,

Thank you for submitting your manuscript to PLOS ONE. After careful consideration, we feel that it has merit but does not fully meet PLOS ONE’s publication criteria as it currently stands. Therefore, we invite you to submit a revised version of the manuscript that addresses the points raised during the review process. Two reviewers have evaluated your submission and identified several aspects that require further clarification. When preparing your revisions, please pay particular attention to including the structural elements requested by Reviewer 1.

We look forward to receiving your revised manuscript.

Kind regards,

Jamie Males

Staff Editor

PLOS ONE

Journal Requirements:

4. We note that Figure 3 in your submission contain [map/satellite] images which may be copyrighted. All PLOS content is published under the Creative Commons Attribution License (CC BY 4.0), which means that the manuscript, images, and Supporting Information files will be freely available online, and any third party is permitted to access, download, copy, distribute, and use these materials in any way, even commercially, with proper attribution. For these reasons, we cannot publish previously copyrighted maps or satellite images created using proprietary data, such as Google software (Google Maps, Street View, and Earth). For more information, see our copyright guidelines: http://journals.plos.org/plosone/s/licenses-and-copyright.

a. You may seek permission from the original copyright holder of Figure 3 to publish the content specifically under the CC BY 4.0 license.

We recommend that you contact the original copyright holder with the Content Permission Form (http://journals.plos.org/plosone/s/file?id=7c09/content-permission-form.pdf) and the following text: “I request permission for the open-access journal PLOS ONE to publish XXX under the Creative Commons Attribution License (CCAL) CC BY 4.0 (http://creativecommons.org/licenses/by/4.0/). Please be aware that this license allows unrestricted use and distribution, even commercially, by third parties. Please reply and provide explicit written permission to publish XXX under a CC BY license and complete the attached form.”

b. If you are unable to obtain permission from the original copyright holder to publish these figures under the CC BY 4.0 license or if the copyright holder’s requirements are incompatible with the CC BY 4.0 license, please either i) remove the figure or ii) supply a replacement figure that complies with the CC BY 4.0 license. Please check copyright information on all replacement figures and update the figure caption with source information. If applicable, please specify in the figure caption text when a figure is similar but not identical to the original image and is therefore for illustrative purposes only. The following resources for replacing copyrighted map figures may be helpful:

USGS National Map Viewer (public domain): http://viewer.nationalmap.gov/viewer/ The Gateway to Astronaut Photography of Earth (public domain): http://eol.jsc.nasa.gov/sseop/clickmap/ Maps at the CIA (public domain): https://www.cia.gov/library/publications/the-world-factbook/index.html and https://www.cia.gov/library/publications/cia-maps-publications/index.html NASA Earth Observatory (public domain): http://earthobservatory.nasa.gov/ Landsat: http://landsat.visibleearth.nasa.gov/ USGS EROS (Earth Resources Observatory and Science (EROS) Center) (public domain): http://eros.usgs.gov/# Natural Earth (public domain): http://www.naturalearthdata.com/

Reviewers' comments:

Reviewer's Responses to Questions

**Comments to the Author**

1. Does the manuscript provide a valid rationale for the proposed study, with clearly identified and justified research questions?

Reviewer #1: Partly

Reviewer #2: Yes

2. Is the protocol technically sound and planned in a manner that will lead to a meaningful outcome and allow testing the stated hypotheses?

Reviewer #1: No

Reviewer #2: Yes

3. Is the methodology feasible and described in sufficient detail to allow the work to be replicable?

Reviewer #1: No

Reviewer #2: Yes

4. Have the authors described where all data underlying the findings will be made available when the study is complete?

Reviewer #1: Yes

Reviewer #2: Yes

5. Is the manuscript presented in an intelligible fashion and written in standard English?

Reviewer #1: Yes

Reviewer #2: Yes

6. Review Comments to the Author

You may also provide optional suggestions and comments to authors that they might find helpful in planning their study.

Reviewer #1: The authors presented the basic concept of the design and enrollment of a cluster-randomized study in Timor-Leste aiming to evaluate the effect of two interventions to improve dietary quality.

The manuscript as submitted does not meet the criteria for publication in PLOS ONE.

(1) The study protocol is not registered in a public accessible database approved by the WHO or ICMJE.

(2) The name of the registry and the trial or study registration number is not included in the abstract.

(3) A SPIRIT schedule of enrollment, interventions, and assessments is not included as the manuscript’s Figure 1.

(4) In general, I would suggest to follow the SPIRIT recommendations step-by-step rigorously throughout the manuscript.

(5) Moreover, including a data management plan and statistical analysis plan should be added as supporting information.

Reviewer #2: Line 106: the sentence seems incomplete.

Line 90: Although the trial design and sample size and sampling procedure are clear, the numbers (n = ) indicated in the CONSORT flow chart are confusing. How many inland villages are there? 24 or 12? How many consumers? 700 or 720?

Line 176: What type of FADs are you considering deploying? The use of wood, concrete or plastic FADs could improve or affect the trial throughout the year. A description of the aggregating devices to be used would be very useful.

7. PLOS authors have the option to publish the peer review history of their article (what does this mean?). If published, this will include your full peer review and any attached files.

Reviewer #1: No

Reviewer #2: No

---

## [Author Response · Author response to Decision Letter 0]

10 Nov 2021

Response to reviewers

We thank the reviewers for their thoughtful comments – they have helped to make the manuscript much stronger. Please see our responses to each comment below. 

Editor:

Response: The manuscript has been formatted accordingly and files have been renamed where necessary according to PLOS One file naming conventions.

Response: The data availability statement is unchanged

Response: The Supporting Information has been deemed unnecessary to include, as it was the consent form, which is standard across studies. It can be supplied upon request

4. We note that Figure 3 in your submission contain [map/satellite] images which may be copyrighted. All PLOS content is published under the Creative Commons Attribution License (CC BY 4.0), which means that the manuscript, images, and Supporting Information files will be freely available online, and any third party is permitted to access, download, copy, distribute, and use these materials in any way, even commercially, with proper attribution. For these reasons, we cannot publish previously copyrighted maps or satellite images created using proprietary data, such as Google software (Google Maps, Street View, and Earth). For more information, see our copyright guidelines: http://journals.plos.org/plosone/s/licenses-and-copyright.

Response: Fig 3 (now Fig 4) was drawn as a vector image by AT so does not require copyright information.

Reviewer 1:

(1) The study protocol is not registered in a public accessible database approved by the WHO or ICMJE.

Response: We have registered the protocol with clinicaltrials.gov, which is an approved database by the WHO. This database is publicly accessible and searchable. 

The link to the clinialtrials.gov registry can be found here (NCT04729829) https://clinicaltrials.gov/ct2/show/NCT04729829?term=NCT04729829&draw=2&rank=1

(2) The name of the registry and the trial or study registration number is not included in the abstract.

Response: This has been added.

(3) A SPIRIT schedule of enrollment, interventions, and assessments is not included as the manuscript’s Figure 1.

Response: Figures 1 and 2 include all of the information about enrolment, interventions, and assessments. We also provide a clear description of the interventions in Table 1. 

(4) In general, I would suggest to follow the SPIRIT recommendations step-by-step rigorously throughout the manuscript.

Response: We used the CONSORT checklist, which is used for reporting randomized controlled trials, and is a standard, detailed tool. 

(5) Moreover, including a data management plan and statistical analysis plan should be added as supporting information.

Response: We have clearly laid out the data management plan and statistical analysis plan in the main text (Sections 2.7 and 2.8). In the mentioned sections, we have explained how variables will be measured, the descriptive analysis, econometrics approach, and mediation analysis. Following the CONSORT checklist tool (S1 Appendix), we have included all of the required information in our manuscript and the corresponding page number. 

Reviewer 2: 

Line 106: the sentence seems incomplete

Response: Thus was a piece of text was accidentally left during the editing process. It has been deleted. 

Line 90: Although the trial design and sample size and sampling procedure are clear, the numbers (n = ) indicated in the CONSORT flow chart are confusing. How many inland villages are there? 24 or 12? How many consumers? 700 or 720?

Response: We agree that this is confusing and we have edited the diagram. Please see the updated version in Figure 1. 

Line 176: What type of FADs are you considering deploying? The use of wood, concrete or plastic FADs could improve or affect the trial throughout the year. A description of the aggregating devices to be used would be very useful.

Response: In the Trial design section, we have added a new description and diagram of the FAD design (Fig 3) and linked to the publication that describe and test the FADs and their deployment methods.

---

## [Decision Letter · Decision Letter 1]

14 Dec 2021

PONE-D-20-40710R1A randomised controlled trial to test the effects of fish aggregating devices and SBC activities promoting fish consumption in Timor-Leste: A study protocolPLOS ONE

Dear Dr. Tilley,

Thank you for submitting your manuscript to PLOS ONE. After careful consideration, we feel that it has merit but does not fully meet PLOS ONE’s publication criteria as it currently stands. Therefore, we invite you to submit a revised version of the manuscript that addresses the points raised during the review process. Specifically, please address reviewer 1's concerns.

We look forward to receiving your revised manuscript.

Kind regards,

Jianhong Zhou

Associate Editor

PLOS ONE

Reviewers' comments:

Reviewer's Responses to Questions

**Comments to the Author**

1. Does the manuscript provide a valid rationale for the proposed study, with clearly identified and justified research questions?

Reviewer #1: Yes

Reviewer #2: Yes

2. Is the protocol technically sound and planned in a manner that will lead to a meaningful outcome and allow testing the stated hypotheses?

Reviewer #1: Partly

Reviewer #2: Yes

3. Is the methodology feasible and described in sufficient detail to allow the work to be replicable?

Reviewer #1: No

Reviewer #2: Yes

4. Have the authors described where all data underlying the findings will be made available when the study is complete?

Reviewer #1: No

Reviewer #2: Yes

5. Is the manuscript presented in an intelligible fashion and written in standard English?

Reviewer #1: Yes

Reviewer #2: Yes

6. Review Comments to the Author

You may also provide optional suggestions and comments to authors that they might find helpful in planning their study.

Reviewer #1: Thank you for the revised manuscript.

PLOS One would not point out in its instructions for authors that the SPIRIT statement is to be used for protocols (https://journals.plos.org/plosone/s/submission-guidelines#loc-guidelines-for-specific-study-types) if CONSORT would be adequate.

The numerous NAs show that CONSORT does not fit in many places. Important aspects in summarizing a study protocol, which are recommend by the SPIRIT checklist, like issues concerning methods against bias, monitoring, ethics or dissemination are not represented at all, i.e. important Information remain withheld from the reader.

I would like to ask you to organize the manuscript along SPIRIT and address all aspects of the checklist.

Reviewer #2: No further explanation or clarification is required for this publication. The comments sent to the authors have been resolved.

7. PLOS authors have the option to publish the peer review history of their article (what does this mean?). If published, this will include your full peer review and any attached files.

Reviewer #1: No

Reviewer #2: No

---

## [Author Response · Author response to Decision Letter 1]

14 Jan 2022

We thank the reviewers for their thoughtful suggestions, and the manuscript has been improved as a result. Please see our responses below. 

Reviewer comment 1: 

The manuscript should describe the methods in sufficient detail to prevent undisclosed flexibility in the experimental procedure or analysis pipeline, including sufficient outcome-neutral conditions (e.g., necessary controls, absence of floor or ceiling effects) to test the proposed hypotheses and a statistical power analysis where applicable. As there may be aspects of the methodology and analysis which can only be refined once the work is undertaken, authors should outline potential assumptions and explicitly describe what aspects of the proposed analyses, if any, are exploratory.

Response: We have specified that control variables to be used in the main estimation equation (1) will come from panel B of Table 2. This helps to minimize the tendency for cherry-picking controls. The way variables are measured could affect the estimated results. Therefore, the methods sections clearly explains how each variable will be measured. We also recognize that the stable unit treatment value assumption (SUTVA) might be violated if other organizations working in the study sites implement interventions similar to those proposed in the study. This is especially problematic if the control group is targeted because of the feeling that they have been left out of a potentially beneficial intervention. We have made efforts to control for this potential problem by collecting data about all organizations working in the study site and the interventions implemented (current and planned). In addition to the randomization, which helps to ensure both treatment and control group participants are equally exposed to the interventions of such organizations at baseline, we discussed with partner organizations and understood their planned future interventions and found that they do not relate to the interventions implemented in our study. Furthermore, we will collaborate with Mercy Corps as our implementation partner to ensure alignment in project interventions and to avoid parallel interventions being targeted at the same respondents by different organizations. Changes have been made in the methodology section accordingly.

Reviewer comment 2: I would like to ask you to organize the manuscript along SPIRIT and address all aspects of the checklist.

Response: This manuscript has been updated and revised accordingly. Please see the supplementary SPIRIT checklist attached.

---

## [Decision Letter · Decision Letter 2]

22 Mar 2022

PONE-D-20-40710R2A randomised controlled trial to test the effects of fish aggregating devices and SBC activities promoting fish consumption in Timor-Leste: A study protocolPLOS ONE

Dear Dr. Tilley,

Thank you for submitting your manuscript to PLOS ONE. After careful consideration, we feel that it has merit but does not fully meet PLOS ONE’s publication criteria as it currently stands. Therefore, we invite you to submit a revised version of the manuscript that addresses the points raised during the review process.

The manuscript has been evaluated by two reviewers, and their comments are available below.

Can you please address the reviewers comments regarding data availability. Please note that to meet PLOS ONE's publication criteria the article must adhere to appropriate reporting guidelines and community standards for data availability (https://journals.plos.org/plosone/s/criteria-for-publication#loc-7). For Registered Report Protocols Authors must confirm that data will be made available upon study completion in keeping with the PLOS data policy (https://journals.plos.org/plosone/s/submission-guidelines#loc-guidelines-for-specific-study-types). 

Could you please carefully revise the manuscript to address all comments raised? Please also carefully review the manuscript for copyediting, and English spelling/grammar. 

We look forward to receiving your revised manuscript.

Kind regards,

Sebastian Shepherd

Associate Editor

PLOS ONE

Journal Requirements:

Reviewers' comments:

Reviewer's Responses to Questions

**Comments to the Author**

1. Does the manuscript provide a valid rationale for the proposed study, with clearly identified and justified research questions?

Reviewer #1: Yes

Reviewer #2: Yes

2. Is the protocol technically sound and planned in a manner that will lead to a meaningful outcome and allow testing the stated hypotheses?

Reviewer #1: Yes

Reviewer #2: Yes

3. Is the methodology feasible and described in sufficient detail to allow the work to be replicable?

Reviewer #1: Yes

Reviewer #2: Yes

4. Have the authors described where all data underlying the findings will be made available when the study is complete?

Reviewer #1: No

Reviewer #2: Yes

5. Is the manuscript presented in an intelligible fashion and written in standard English?

Reviewer #1: Yes

Reviewer #2: Yes

6. Review Comments to the Author

You may also provide optional suggestions and comments to authors that they might find helpful in planning their study.

Reviewer #1: Please add where all data underlying the findings will be made available when the study is complete (see 4.). All my other comments have been satisfactorily addressed.

Reviewer #2: No further explanation or clarification is required for this publication. The comments sent to the authors have been resolved.

7. PLOS authors have the option to publish the peer review history of their article (what does this mean?). If published, this will include your full peer review and any attached files.

Reviewer #1: No

Reviewer #2: No

---

## [Author Response · Author response to Decision Letter 2]

4 Apr 2022

In response to Reviewer #1’s comment: “Please add where all data underlying the findings will be made available when the study is complete (see 4.). All my other comments have been satisfactorily addressed.” We have added the following text to clarify where the data will be made available online (lines 257-262):

"Identifiable survey data will be archived and accessible to only the research team and those approved by the entire research team. Anonymised data from the study will be published on the Harvard Dataverse (https://doi.org/10.7910/DVN/VZJYIZ) to facilitate future research such as pooled analysis with other studies for purposes of meta-analysis or multi-site analysis. Anonymous fisheries landings data are publicly available on Harvard Dataverse (https://dataverse.harvard.edu/dataverse/peskas) and updated automatically every month."

---

## [Editor Report · Decision Letter 3]

18 May 2022

A randomised controlled trial to test the effects of fish aggregating devices (FADs) and SBC activities promoting fish consumption in Timor-Leste: A study protocol

PONE-D-20-40710R3

Dear Dr. Tilley,

We’re pleased to inform you that your manuscript has been judged scientifically suitable for publication and will be formally accepted for publication once it meets all outstanding technical requirements.

Kind regards,

George Vousden

Staff Editor

PLOS ONE
---

## [Editor Report · Acceptance letter]

17 Jun 2022

PONE-D-20-40710R3 

A randomised controlled trial to test the effects of fish aggregating devices (FADs) and SBC activities promoting fish consumption in Timor-Leste: A study protocol 

Dear Dr. Tilley:

I'm pleased to inform you that your manuscript has been deemed suitable for publication in PLOS ONE. Congratulations! Your manuscript is now with our production department. 

Kind regards, 

on behalf of

Dr. George Vousden 

Staff Editor

PLOS ONE